# A Rapid Bioassay Test for Assessing Environmental Contamination Using the Marine Sedentary Polychaete *Hydroides elegans*

Priya Sivakumar [1,†], Gomathi Srinivasan [1,†], Madhuvandhi Janardhanam [1], Rekha Sivakumar [1], Priscilla Niranjani Marcus [1], Sujatha Balasubramaniam [1], Gopalakrishnan Singaram [2] and Thilagam Harikrishnan [1,*]

[1] Postgraduate and Research Department of Zoology, Pachaiyappa's College for Men, Chennai 600030, India; priyasivakumar400@gmail.com (P.S.); gomathikarthick18@gmail.com (G.S.); madhujanardhanam@gmail.com (M.J.); rekhasivakumar698@gmail.com (R.S.); priscillaniranjani@gmail.com (P.N.M.); lbsujatha@gmail.com (S.B.)

[2] eProov Assessment Solution PVT Limited, Chennai 600035, India; gopalthilagam@gmail.com

[*] Correspondence: thilagampachaiyappas@gmail.com

[†] These authors contributed equally to this work.

**Abstract:** To investigate the impact of environmental contaminants on the early life stages of the marine polychaetae *Hydroides elegans*, a toxicity test was designed. In our previous study, we reported gametes and embryos of *H. elegans* were sensitive to heavy metal pollution and effluents. In continuation of this, we used *H. elegans* gametes to assess the water quality of samples taken along the southeast coast of India. The samples were collected from five different locations of the Chennai coast (Muttu Kadu, Neelangarai, Marina, Royapuram, and Ennore), and two different bioassay toxicity tests were performed. Sperm and eggs were pre-exposed to water samples taken from different locations to assess the water quality. Water samples collected from Ennore station and the Royapuram fish landing center were found to be more polluted than those collected from other locations. Sperm were shown to be more sensitive than eggs. The different morphological effects produced by water samples reflected the defects in the early differentiation of embryonic cells. Since fertilization can be inhibited in the presence of any xenobiotic, both fertilization and early development could be used as a biological indicator for a rapid bioassay to monitor marine pollution. The percentage of successful fertilization and early development was comparatively higher at the reference site (Neelangarai) and in the seawater samples collected from Marina. The physicochemical characteristics of the seawater from these sampling stations corroborated the findings of this investigation. Our results showed that *H. elegans* gametes were highly sensitive to any contaminant present in the seawater, and confirmed previous findings that this polychaetae can be routinely used as a test organism for ecotoxicological bioassays in tropical and subtropical regions.

**Keywords:** rapid bioassay; gametes; embryos; *Hydroides elegans*

## 1. Introduction

Pollution associated with urbanization threatens the water quality and has been identified as one of the most serious problems affecting coastal and marine flora and fauna [1]. Therefore, it is very important to develop methods to monitor pollutants that pose risks to humans and biota [2]. Biotoxicity tests are simple laboratory bioassays that have the advantage of reflecting the total toxicity of pollutants in a complex coastal and marine environment. Bioassays enable such detection by evaluating the biological response of marine organisms, particularly at highly sensitive early life stages. Several bioassay testing protocols have been developed and are widely employed to assess the toxicity of contaminants using oysters, mussels, sea urchins, sand dollars, fishes, and mysids as test

organisms [3–9]. However, this test is commonly used in Western countries for effluent monitoring programs, and it requires several hours or days in achieving the outcome. Moreover, these toxicity tests are based on field-collected organisms, some of which are viable gametes at a prescribed period, limiting their application in routine toxicity testing.

*Hydroides elegans,* a sedentary polychaete prevalent in temperate regions, produces viable gametes throughout the year [10]. A previous study employing *H. elegans* embryos and larvae demonstrated that heavy metal exposure and effluents caused fertilization obstruction and developmental arrest [11,12]. Although several studies on *H. elegans* species are available, less attention has been paid to the use of *H. elegans* gametes (eggs and sperm) in evaluating the effects of pollutants in seawater. Therefore, the objectives of this study were to determine the suitability of *H. elegans* gametes as potential components of bioassays by assessing their sensitivity to seawater obtained from the east coast of South India. The advantage of employing these animals for toxicological investigations is that they can be quickly induced to release germ cells, which has considerable potential for application in laboratory toxicity testing. To the best of our knowledge, this is the first study in Asia to investigate the effectiveness of marine polychaetae gametes as possible bioindicators of aquatic pollution. This research might aid in the assessment of coastal water quality by elucidating embryonic development stage deformities in a possible risk group of faunal populations inhabiting a similar environment.

## 2. Material and Methods

### 2.1. Study Area Description

In the present study, five sampling locations: station 1 (Ennore 13.13′50° N, 80.19′55° E), station 2 (Royapuram fishing harbor 13.07′45° N, 80.17′58° E), station 3 (Marina 13.03′37° N, 80.17′14° E), station 4 (Neelangarai 12.56′54° N, 80.15′42° E), and station 5 (Muttu Kadu backwater 12.48′27° N, 80.14′54° E) were selected for seawater collection and analysis of metal and physicochemical parameters (Figure 1). The seawater samples were collected 50 m off the coast of the sampling stations at a depth of 0.5 m. The samples were filtered through a pre-weighed standard glass-fiber filter (0.45 μm, Millipore GF/C) to remove suspended solids larger than 10 microns. The seawater sample collected from station 4 (Neelangarai) was used to culture and maintain organisms, and served as the reference water. For the toxicity test, filtered seawater was aerated continuously to keep dissolved oxygen levels >5 mg/L [13]. Water samples collected from five sampling stations were analyzed for metals and physicochemical parameters (Table 1) [14].

**Table 1.** Physicochemical parameters of the water samples collected from the East coast of India.

| Parameters | Ennore | Royapuram | Marina | Neelangarai | Muttu Kadu |
|---|---|---|---|---|---|
| **Physicochemical Parameters** | | | | | |
| WT (°C) | 25.50 ± 3.69 | 26 ± 2.70 | 26.33 ± 1.24 | 25.83 ± 1.43 | 27.33 ± 1.40 |
| pH | 8.07 ± 0.18 | 7.90 ± 0.32 | 8.10 ± 0.21 | 8.06 ± 0.18 | 8.15 ± 0.23 |
| Oxygen (mg/L) | 4.50 ± 0.78 | 4.25 ± 0.68 | 4.80 ± 0.24 | 4.83 ± 0.49 | 4.90 ± 0.28 |
| Salinity (PSU) | 33.50 ± 5.80 | 31.25 ± 4.34 | 32.66 ± 1.88 | 34.33 ± 0.47 | 33.06 ± 1.08 |
| **Heavy Metals (mg/L)** | | | | | |
| Copper | 0.022 ± 0.007 | 0.015 ± 0.004 | 0.012 ± 0.002 | 0.007 ± 0.002 | 0.011 ± 0.003 |
| Cadmium | 0.016 ± 0.004 | 0.015 ± 0.002 | 0.007 ± 0.001 | 0.003 ± 0.001 | 0.008 ± 0.002 |
| Cobalt | 0.024 ± 0.003 | 0.012 ± 0.003 | 0.006 ± 0.001 | 0.002 ± 0.001 | 0.009 ± 0.001 |
| Chromium | 0.027 ± 0.002 | 0.014 ± 0.003 | 0.009 ± 0.001 | 0.009 ± 0.002 | 0.011 ± 0.001 |
| Lead | 0.023 ± 0.001 | 0.028 ± 0.004 | 0.010 ± 0.001 | 0.007 ± 0.004 | 0.012 ± 0.003 |
| Zinc | 0.039 ± 0.003 | 0.019 ± 0.005 | 0.011 ± 0.004 | 0.006 ± 0.005 | 0.014 ± 0.003 |
| Nickel | 0.016 ± 0.002 | 0.022 ± 0.002 | 0.019 ± 0.001 | 0.009 ± 0.002 | 0.015 ± 0.002 |
| Mercury | 0.012 ± 0.001 | 0.004 ± 0.001 | BDL | BDL | BDL |
| Iron | 0.191 ± 0.026 | 0.139 ± 0.016 | 0.059 ± 0.025 | 0.088 ± 0.042 | 0.119 ± 0.021 |
| **Hydrocarbon (μg/L)** | | | | | |
| TPH | 17.34 ± 3.96 | 11.14 ± 2.38 | BDL | BDL | 3.21 ± 2.09 |

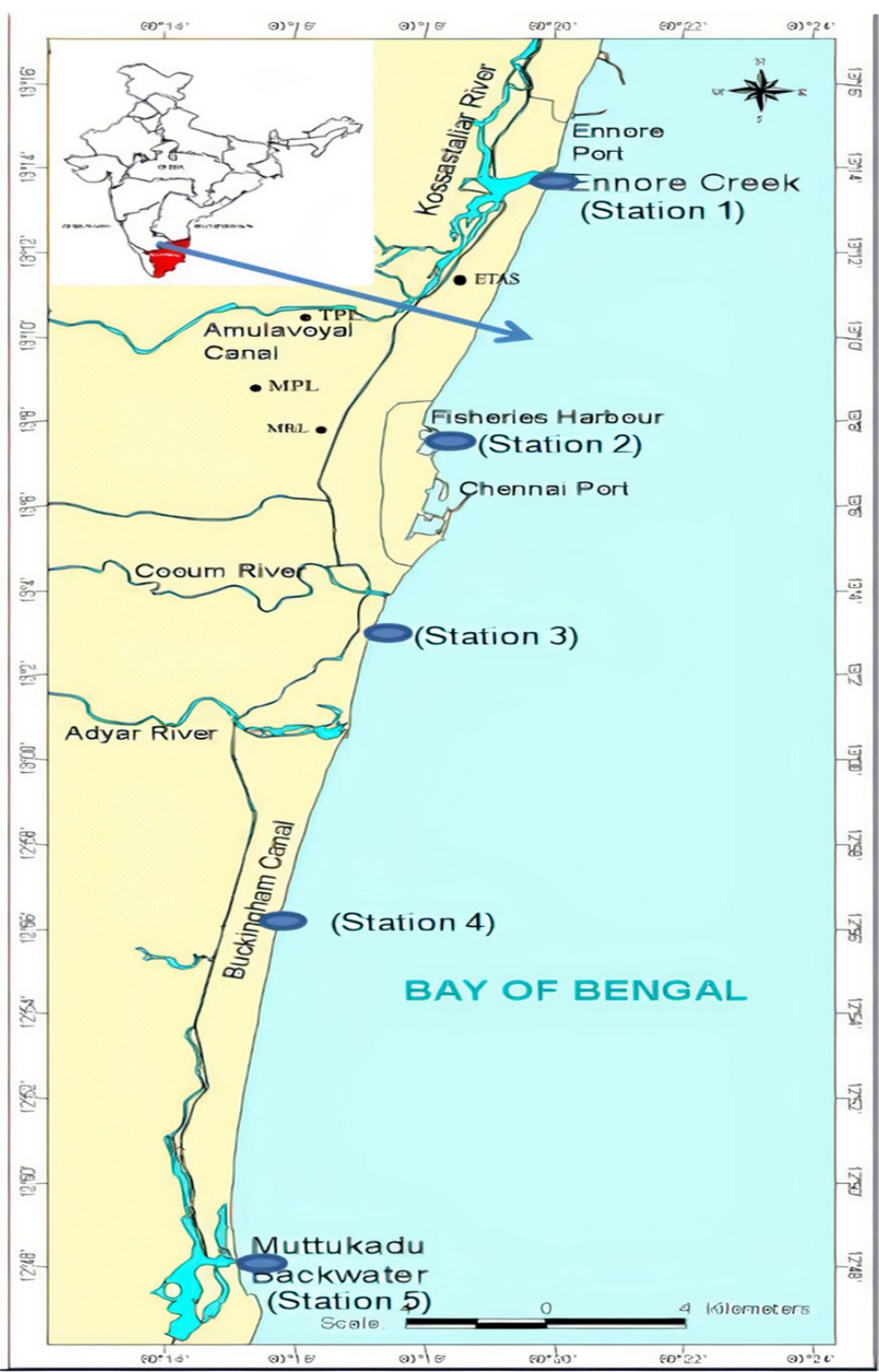

**Figure 1.** Map showing location of sampling sites along the southeast coast of India.

*2.2. Collection of Specimen*

Specimens of *Hydroides elegans*, a marine polychaetae worm, were collected from nylon ropes suspended from a floating pier at Pulikat Lake and transported to the laboratory. The *H. elegans* were kept in a glass aquarium with seawater and fed with fresh cultures of the phytoplankton *Isochrysis galbana*. Polychaetae worms were acclimatized to laboratory conditions for a week; during acclimatization, the seawater was changed daily. Dissolved oxygen ($7 \pm 2$ mg/L), salinity ($34 \pm 1$ ppt), temperature ($28 \pm 1$ °C), and pH ($8.1 \pm 0.1$)

were maintained in the glass aquarium, and the light was provided in a light–dark cycle of about 14:10 h [11].

The tests were carried out in 50 mL acid-washed glass beakers. Preparation of gametes was performed as described by Gopalakrishnan et al., [10]. Fertilization was confirmed by seeing an elevation of the fertilization membrane within 5 to 7 min after mixing of gametes, and a visible cell division was used to determine the embryo development. Unfertilized eggs or cells that had not been differentiated were considered undeveloped eggs. The experiment was repeated three times and maintained in triplicates to ensure that the results were consistent on each occasion. Three data points for each sampling station were combined to give the average and standard deviation (SD) of nine samples (3 triplicate × 3 repetitions), which were then statistically analyzed. Though blastula stage was reached within 2 h in the reference water, the experiment was terminated after 3 h, or when most of the embryos reached the blastula stage, by adding 10% neutral buffered formalin prepared in seawater, and the success rate of the respective embryonic stage was confirmed by counting the first 100 embryos encountered and grading them as developed or undeveloped [10,11]. Photomicrographs were taken using a Pentax K1000 (Tokyo, Japan) camera mounted to a Weswox microscope.

### 2.3. Heavy Metal Analysis

The chemicals used were analytical-grade reagents, and Milli-Q (Elix UV5 and Milli-Q, Millipore, Burlington, MA, USA) water was used for solution preparation. Before sample preparation, the Teflon vessel and polypropylene containers were cleaned by soaking in 5% $HNO_3$ for 24 h, then rinsed with Milli-Q water and dried. Metal analysis was performed using an air–acetylene flame atomic absorption spectrophotometer (AAS: Spectra AA-10 Varian, Victoria, Australia), as described by Tewari et al., [15] According to Weltz and Schubert-Jacobs [16], mercury was analyzed in seawater using automated cold vapor AAS. To check for metal recovery, quality control samples made from standard heavy metal solutions were analyzed once for every three samples. The percentages of recovery were greater than 96% (IAEA shrimp, MA-A-3/TM; and IAEA simulated freshwater, W-4). The measured values were in good agreement with the certified values (<10% deviation).

### 2.4. Total Polyaromatic Hydrocarbon (TPH)

The extraction procedure was carried out following Adeniji et al., [17]. The 2 L of each collected water sample was vacuum-filtered through fiberglass filters (Whatman GF/F) with nominal pore sizes of 0.7 μm and twice extracted in a separation funnel with 50 mL of n-hexane. The 100 mL extracts were dried over sodium sulphate before being reduced to 5 mL in a rotary evaporator. These final extracts were used to analyze TPH using UV–fluorescence spectrometry (Perkin Elmer LS 30 UV Spectrometer, Billerica, Massachusetts, MA, USA) at 310 nm/360 nm excitation/emission wavelengths and with chrysene solution used as a quantitative standard.

### 2.5. Exposure of Eggs to Different Seawater Samples (Egg Cell Bioassay)

About 200 eggs were placed in glass beakers containing 50 mL of seawater collected from various stations. Before being used in the experiment, the eggs were exposed to such sampling water for 30 min. Then, we added 100 μL of untreated sperm suspension to the exposed eggs. Fertilization was checked 5 min after sperm and oocytes were mixed and continued for 3 h until the oocytes had undergone fertilization and embryonic development and/or disintegrated. After 3 h, a few drops of 20% buffered formalin was added to all beakers, and the fertilization/developmental rate was calculated as mentioned above.

### 2.6. Exposure of Sperm to Different Seawater Sample (Sperm Cell Bioassay)

Approximately 500 μL of diluted sperm (40,000 sperm μL$^{-1}$) were added to the beaker containing 50 mL of seawater collected from various stations. Before being used for insemination, sperm were exposed for 30 min to such sampling water. About 200 numbers

of untreated eggs were added to the pre-exposed sperm. As mentioned above, fertilization success was checked after 5 min of mixing of gametes, and the remaining procedure and the endpoints were similar to those mentioned above (Section 2.5).

### 2.7. Statistical Analysis

SPSS software (version 20, IBM, Armonk, NY, USA) was used to calculate the statistical outcome. The significance of the results was determined by calculating the mean and standard deviation of nine observations per group. For all experiments, a one-way analysis of variance (ANOVA) was used to compare the significant differences between the sampling stations. The Tukey test was used in cases where the null hypothesis of the homogeneity of the effect of the factor value had been rejected. The contaminants and bioassay variables were normalized to render the data dimensionless for factor analysis (FA) [18,19]. The factor loadings were sorted according to the criteria of Liu et al. (2003); i.e., strong, moderate, and weak, corresponding to absolute loading values of >0.75, 0.75–0.50, and 0.50–0.30, respectively. The Kaiser–Meyer–Olkin (KMO) criterion was followed to determine sampling adequacy.

## 3. Results

### 3.1. Exposure of Eggs to Different Seawater Sample

Fertilization success significantly differed among the water samples collected from different stations. About 42% of the eggs were inhibited from fertilization when they were exposed to the station 1 (Ennore) sample. The fertilization rate decreased significantly ($p < 0.05$) when the experiment was conducted in the samples collected from stations 1, 2 and 5. Normal embryonic development without any deformity was observed in the reference water sample (Figure 2b–e). The percentage of fertilization success was significantly higher in the samples obtained at station 3 (Marina, $85 \pm 1.34$) and station 4 (Neelangarai, $90.33 \pm 0.61$). Similarly, station 5 (Muttu Kadu) and station 2 (Royapuram) showed $78.50 \pm 0.88\%$ and $70.0 \pm 1.43\%$ fertilization success, respectively (Figure 3a). When eggs were exposed to seawater samples taken from stations 1 and 2, the proportion of eggs that reached the two-cell stage following fertilization was significantly lowered compared with the reference site (Figure 3b). The success rate of four-cell stage embryos was reduced to 32% and 23%, respectively (Figure 3c) when eggs were exposed to seawater samples collected from stations 2 and 1, respectively. However, when the eggs were placed in the reference seawater, around 60% of the fertilized embryos reached the blastogenic stage. Contrary to this, when the eggs were exposed to seawater collected from station 1, only 18% of the embryos reached the blastogenic stage (Figure 3d). Abnormal development of the embryo was higher in the eggs treated with station 1 and station 2 seawater samples (Figure 2f). All values were statistically significant compared with the reference control points (Table 2).

**Table 2.** ANOVA for FM and blastula stage (egg and sperm bioassays).

| ANOVA Table | SS | DF | MS | F | *p*-Value (Sig.) |
|---|---|---|---|---|---|
| Treatment (between columns) Egg bioassay (FM-stage) | 3912 | 4 | 978.0 | 138.3 | 0.0001 |
| Treatment (between columns) Sperm bioassay (FM stage) | 8538 | 4 | 2134 | 121.3 | 0.0001 |
| Treatment (between columns) Egg bioassay (blastula stage) | 5970 | 4 | 1493 | 62.14 | 0.0001 |
| Treatment (between columns) Sperm bioassay (Blastula stage) | 2360 | 4 | 589.9 | 66.93 | 0.0001 |

BDL: below detectable limit. Values are expressed as mean $\pm$ SD.

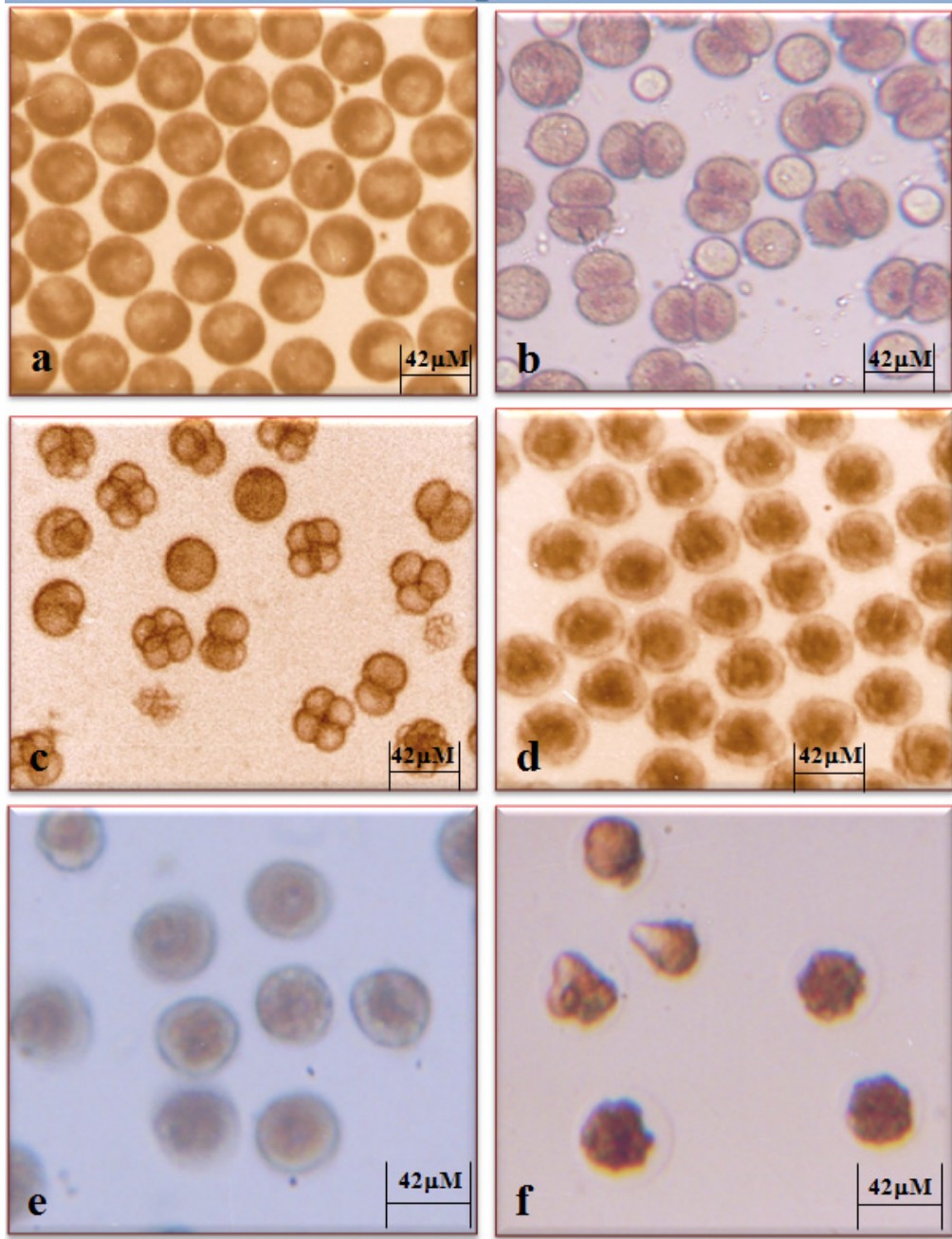

**Figure 2.** Delayed and abnormal development of embryos of *H. elegans* in sample water collected from different regions of southeast coast of India: (**a**) eggs before fertilization; (**b**) fertilized eggs and two-cell stage; (**c**) four-cell stage; (**d**) blastula stage; (**e**) deformed blastula stage when the eggs were exposed to station 2 (Royapuram) seawater; (**f**) deformed embryonic development when the sperm were exposed to station 1 (Ennore) seawater.

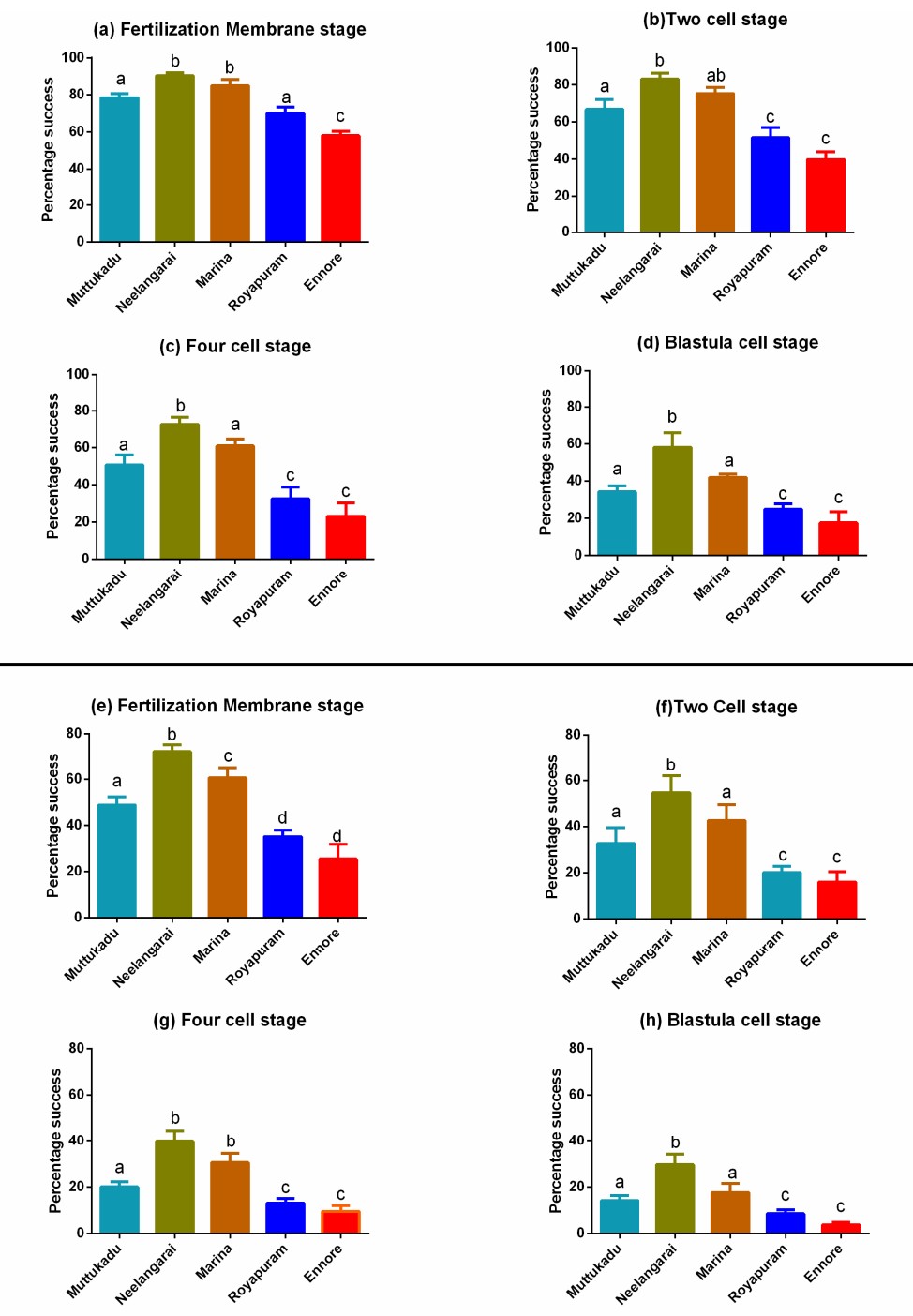

**Figure 3.** Percent successful development of different embryonic development stages of *H. elegans*. (**a–d**) Egg cell bioassay. (**e–f**) Sperm cell bioassay. Each bar represents the mean ± SD. Two-way analysis of variance followed by Tukey's post hoc test were performed. Different letters denote significant differences between sampling stations, whereas the same letter denotes no statistically significant differences ($p < 0.05$) between different sampling stations.

### 3.2. Exposure of Sperm to Different Seawater Samples

Although the experiment lasted only three hours, the sperm bioassay yielded substantial results. Within this period, eggs that did not fertilize and divide were considered abnormal, dead, and disintegrated. Fertilization success rates differed dramatically across water samples obtained at different locations, with seawater samples from station 1 (Ennore) and station 2 (Royapuram) showing a considerable decrease in fertilization success

rates (25.50 ± 2.59% and 35.17 ± 1.10%, respectively) (Figure 3e). Fertilization rates were significantly greater (*p* < 0.05) when experiments were conducted in seawater obtained from the reference site (Neelangarai, 72.33 ± 1.22%), and the same significantly decreased when the experiments were conducted in the samples taken from station 5 (Muttu Kadu, 48.83±1.44%). Similarly, the percentage of embryos developed into two-cell and four-cell stages were decreased considerably in comparison to the reference station (Neelangarai, Figure 3f,g). When the experiment was conducted in the seawater samples taken from station 1 (Ennore), the blastogenic success was only 3%. Most of the embryonic development was abnormal (Figure 2g). When compared to the reference station (station 4), the blastogenic success was significantly lower in all the stations.

The metal concentration in the seawater and the physicochemical properties of the seawater are presented in Table 1. The background concentrations of the heavy metals were relatively high in the seawater samples collected from stations 1 and 2. The TPH concentration was found to be higher at both stations 1 and 2. However, at stations 3 and 4, the TPH concentration was below the detectable level.

In the PCA analysis, the first two principal components accounted for 82.32% of the overall variance of the data. The percentages of variance explained by different principal components were 56.04% and 26.28%. The rotated component matrix developed by a PCA of the bioassay of sampling sites is given in Table 2. Based upon the scatter plot of the first two principal components of the bioassay, the pattern in relative magnitudes of parameters was similar among locations. Specifically, station 1 and station 2 had the most similarity, followed by station 3 (Figure 4).

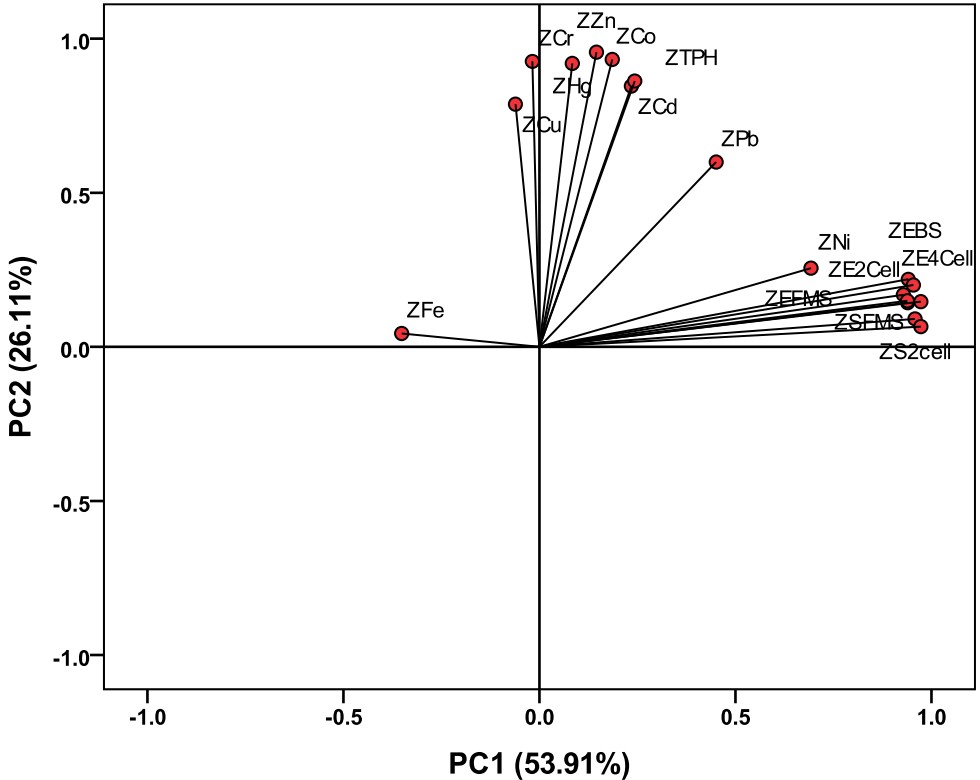

**Figure 4.** Scatter plot of (A) contaminants, bioassay, and ordinates along with the first two principal components of the different bioassay values for the southeast coastal regions of India.

## 4. Discussion

Oceanic and coastal species produce a large number of small eggs that hatch early in development and have a long larval phase, allowing for widespread dispersal [20,21], and their survival may be affected by anthropogenic contaminants. Contaminants in the environment have been proven to affect not just the function, distribution, and/or community

structure of marine species, but also their biological characteristics [22]. This necessitated the development of a bioassay test using *H. elegans* (a sedentary marine polychaetae), which has been advocated as a suitable organism for toxicity test of contaminants [10–12]. The present study proved that the gametes of *H. elegans* can be used as a bioassay tool for basic toxicological studies in coastal waters.

The physicochemical characteristic of seawater in samples collected at various sites is shown in Table 1. The data suggest that water samples collected from station 1(Ennore) were highly polluted, followed by station 2 (Royapuram fish landing center), station 5 (Muttu Kadu), and station 3 (Marina). The seawater quality of these locations worsened the developmental phases of marine species, as indicated by sperm and egg cell bioassays, and as evidenced by a prior study showing Ennore Creek was more polluted, and the fish landing center was next [12]. When gametes were exposed to polluted seawater samples before fertilization, the development of the gametes was impaired. The sperm were found to be more sensitive than the eggs. Even if the eggs were fertilized and continued to develop despite gametes being exposed to seawater previously, the proportion of developmental embryos was drastically decreased, or most of the developed embryos were malformed or disintegrated. Development of *H. elegans* embryos in water samples collected on the east coast of southern India revealed poor water quality at stations 1 (Ennore) and station 2 (Royapuram fishing harbor), with considerable spatial variations. The seawater samples from station 3 (Marina) and station 5 (Muttu Kadu) were not quantitatively hazardous to *H. elegans* embryos, but their rate of development was much slower, and the cause of these changes is unknown. The waters of Ennore and Royapuram are open systems that are directly influenced by tidal impulses, regardless of plausible reasons. If water quality is not maintained at a high enough level, nearby marine habitats may be polluted. Pollution levels at stations 1 and 2 seawater increased *H. elegans* embryo deformities, suggesting that the water quality in these regions may not be suitable for sensitive marine invertebrate species in the near future.

Exposure time is a critical factor in the early development of embryos; as gametes' exposure time to contamination rises, fertilization success drops rapidly. The natural decline in sperm viability with time in diluted media is a complicating factor that may generate spurious toxicity in these types of research. Thus, the current study demonstrated that exposing gametes before fertilization reduced the fertilization rate in *H. elegans*, indicating that the fertilization membrane generated may prevent contaminants from accessing the oocytes. According to Barron et al. [23] soluble and fixed reactive sulfhydryl (–SH) groups influence sperm activation and motility, and increased metal levels in seawater can impact the sulfhydryl groups [4]. Because fixed sulfhydryl groups are key components of enzyme reactions that govern respiration, high metal levels might contaminate the respiratory and metabolic pathways [23].

The multivariate analysis helped elucidate relationships among different stages, the bioassays, and sampling stations. The five stations along the South India Coast exhibited diverse patterns of contaminants, ranging from less- to more-polluted regions. A PCA performed on the sperm cell and egg cell bioassays proved to be an effective method to distinguish among locations. A factor analysis rendered three significant factors (Eigenvalue > 1) that explained 87.455% of the total variance of the dataset. The KMO criterion (0.719) indicated that through FA, a significant reduction in the dimensionality of the original data set could be achieved [24]. The factor loadings and communality of contaminants and bioassay variables are presented in Table S1 (Supplementary Materials). Factor 1 (VF1) explained 53.909% of the total variance, and indicated a strong positive loading for the sperm bioassay followed by the egg bioassay, and all the values were significant and higher than 0.92 (Table S1 and Figure 4). This could be attributed to the diverse impact of human activities in the surrounding vicinities. Factor 2 (VF2) explained 26.112% of the total variance, and indicated a strong positive loading for heavy metals and the PAH (Table S1 and Figure 4). This could be attributed to the delay in development and deformities of the embryos due to the presence of high levels of contaminants in the coastal regions.

Thus, using the first principal component, the five sample sites were ranked as follows: Stn 1 > Stn 2 > Stn 5 > Stn 3 > Stn 4. Since sperm were reported to be more sensitive to organic and inorganic pollutants, the presence of a high level of contaminants at Stn 1 and Stn 2 were more likely to have affected the sperm cells. Our previous study found that Stn 1 was more polluted [12]. Thus, this area may not be suitable for marine invertebrate culture and development in the coming years if anthropogenic contaminant discharge is not mitigated in an environmentally sustainable manner.

As evidenced by our previous study, when the gametes were mixed, fertilization success was greater [11], and a delay in mixing of gametes or pre-exposure of gametes to contaminants could lead to deformities during embryonic development. The current study's findings revealed that even small quantities of metals or PAHs had a synergistic influence on the effectiveness of fertilization in the marine animals. This polychaetae's early development is analogous to that of sea urchins in its early stages. A contaminant response relationship could be established using abnormality and death as an outcome in a toxicity test. Our findings showed that *H. elegans'* early development was very sensitive to any organic or inorganic pollutant, and that this polychaetae may be used as a test organism for both acute and chronic ecotoxicity bioassays in tropical environments. Further, the bioassay utilizing the gametes of *H. elegans* could be used in the future to identify the suitability of seawater quality for larval rearing in aquaculture.

**Supplementary Materials:** The following supporting information can be downloaded at: https://www.mdpi.com/article/10.3390/w14111713/s1, Table S1: Varimax-rotated factor loadings and communality of contaminants and bioassay tests.

**Author Contributions:** P.S. and R.S.: Experimental analysis and sample collection processing; G.S. (Gomathi Srinivasan): Experimental analysis; M.J. and P.N.M.: Sample collection processing; S.B.: Data analysis and manuscript preparation; G.S. (Gopalakrishnan Singaram): Data analysis and interpretation; T.H.: Experimental design, reviewing of data and writing of the original manuscript. All authors have read and agreed to the published version of the manuscript.

**Funding:** No funding was received for this study.

**Institutional Review Board Statement:** For the care and use of animals, all applicable national, and/or institutional guidelines as per the Animal Use Committee of India were followed.

**Informed Consent Statement:** Not applicable.

**Data Availability Statement:** The original contributions presented in the study are included in the article/Supplementary material; further inquiries can be directed to the corresponding author/s.

**Acknowledgments:** The authors thank the head of the Research Department of Zoology and the principal of Pachaiyappa's College for Men, Chennai 600 030, India, for extending their support throughout the study period and experiment.

**Conflicts of Interest:** The authors declare no conflict of interests.

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
