# Peer review of "A Rapid Bioassay Test for Assessing Environmental Contamination Using the Marine Sedentary Polychaete Hydroides elegans"

_water, doi:10.3390/w14111713_

Round 1

Reviewer 1 Report

The topic of the work is quite relevant. Express analysis of marine biotopes using an embryo test is widely used in ecology. Sea urchin gametes and embryos are commonly used because they can easily produce gametes and embryos during the long breeding season.

("Sea Urchin Embryo Test" (Kobayashi N., 1984. Marine ecotoxicological testing with Echinoderms // Personne G., Jaspers E., Clans C. (eds.) Ecotoxicological Testing for the Marine Environment. Belgium, Breden: State Univ. Ghent & fast Mar. Sci. Res. V. 1, p. 341 - 405)

Beiras, R., Durán, I., Bellas, J., and Sánchez‐Marín, P. 2012. Biological effects of contaminants: Paracentrotus lividus sea urchin embryo test with marine sediment elutriates. ICES Techniques in Marine Environmental Sciences No. 51. 13 pp

LilianaSaco-Álvareza, IriaDurána, J.Ignacio Lorenzoa, RicardoBeirasab Methodological basis for the optimization of a marine sea-urchin embryo test (SET) for the ecological assessment of coastal water quality. Ecotoxicology and Environmental Safety. V. 73, 4, 2010, P. 491-499. )

The Indian authors of the work under review suggest using gametes and early embryos of the sessile polychaete Hydroides elegans as an embryo test. The authors showed that polychaete spermatozoa are more sensitive to toxicants than eggs and embryos. This is consistent with the results of other authors. Testing the waters of different bays on the east coast of South India is of practical importance for the local population.

Reviewer 2 Report

The manuscript titled “ a rapid bioassay test for assessing environmental contamination using the marine sedentary polychaete hydroides elegans” presents an interesting approach to examine the impact of pollutant in the aquatic environment.

The manuscript can be considered but with major revision.

  1. The abstract is not well written, the idea of the experiment on what was done was not clearly stated
  2. “ using gametes from hydroides elegans, a toxicity test was carried out to establish the environmental impact on a marine sedentary polychaete.” What environmental impact of what?
  3. The general text should be revied and edited by a native english speaker
  4. There is no line number, making it difficult to make reference to corrections in text
  5. The figure 1 should be separated, placing sample site under the description of study area, and the egg fertilization and development should be placed where appropriate
  6. “polychaete worms acclimatized to laboratory conditions for a week and feeding regime.” This sentence is not complete a description of how the fertilization was carried out should be described earlier before talking about what how the fertilized egg was exposed, moreso, the figures on egg development should be placed after this
  7. “fertilization success rates are significant when eggs are exposed to seawater collected at the reference site, whereas approximately 42% of eggs inhibit fertilization when exposed to ennore station seawater samples”…. The results description needs to be rewritten
  8. The use of the word egg and sperm toc=xicity gives the idea that the egg and sperm are pollutants themselves causing toxicity to something. The right construction has to be used.
  9. If you were trying to describe fertilization success rate, then this statement “fertilization rates were considerably lowered (p<0.05) when experiments were conducted on seawater obtained from stations 4 (neelangarai) 72.33±1.22% and station 5 (muttu kadu) 48.83±1.44%. Is wrong. Because station 4 had the lowest pollution and it is the one you refer to as reference station, right?
  10. What is the magnification of the images of the egg development? Please include
  11. There are lots of repetitions in the discussion

Reviewer 3 Report

The manuscript entitled “A Rapid Bioassay Test for Assessing Environmental Contamination Using the Marine Sedentary Polychaete Hydroides elegans” by Priya et al. reported their research work on using Hydroides elegans for water contamination rapid assessment. The idea for using elegans as rapid indicator of water contamination is interesting and may draw attention from researchers in the related area.

However, the design of the experiment is not convincing. As the author presented in the conclusion “The current study's findings reveal that even small quantities of metals or PAHs have a synergistic influence on the effectiveness of fertilization in marine animals”, the authors should design single-contaminant-polluted samples and perform experiments, rather than using real multiple-contaminants-polluted samples in the study.

Reviewer 4 Report

The authors of this studies have performed experiment to develop a rapid bioassay test to define the toxicity of different sample of coastal water. In this study, it was established that some samples were toxic toward the egg development as well as the male gametes. However, before considering this work for publication some comment have to be addressed.

1/ The first main comment is the quality of the abstract that to be improved. While reading it the work perform in this study appears foggy. Please reshape the abstract to allow to have a clear vision of the main results and conclusions.

2/ A strong effort need to be done regarding the figure and table of the paper. It is not clear why there is picture of different stage of eggs together with the picture of the sampling location. They have to be in two different figure and each figure has to be used.

 3/ Same issue for table 1. The table 1A make sense but Table1B is really confusing. Please improve the statistic and explain them better.

4/ In the material and method please indicate the GPS coordinate of the sampling points.

5/ The authors have incubated the eggs and sperm 30 min before doing the fertility tests. But there is no clear explanation of how this fertility test is conducted. Does the authors are keeping the eggs + sperm in the egg exposed water up to 3 hours?  In this case the toxicity is may be due to this 3 h incubation and not linked to the 30 min. 

6/ After the 30 min of incubation of the sperm does the authors have investigated the physiological parameter? It can be even more sensitive and faster.

7/ What do the authors mean here " The seawater samples were collected from 50 m off the coast of the Bay of Bengal at the depth of 0.5 m. The samples were filtered through a preweighed standard glass-fiber filter (0.45 μm Millipore GF/C) to remove suspended solids larger than 10 microns. The same seawater was used to culture and maintain organisms, as well as serve as reference water " ??

8/A PCA between the toxicological effects and the water parameters would bring an interesting representation of the driving parameter in the toxicity. 

Round 2

Reviewer 1 Report

It seems manuscript may publish in this variant.

Author Response

Thanks for reviewer 

Reviewer 2 Report

I appreciate the work done so far, but i still feel that the term egg toxicity can be changed to embryotoxicity to complement spermiotoxicity. i will suggest not to use he terms sperm toxicity and egg toxicity. 

This statement "sperm toxicity was shown to be more sensitive than egg toxicity." can be better be said as. ' the result shows that the sampled water was more toxic to the sperm than the egg, or the sperm was more sensitive to the pollutants in sampled water they were exposed to than the eggs.

because if you say egg toxicity or sperm toxicity, it means the egg or sperm is the substance that is causing toxicity to something.

what you are studying is the toxicity of the sampled waters to egg and sperm of the animal.

Author Response

Comments and Suggestions for Authors

Comments:  I appreciate the work done so far, but i still feel that the term egg toxicity can be changed to embryotoxicity to complement spermiotoxicity. i will suggest not to use the terms sperm toxicity and egg toxicity. 

 This statement "sperm toxicity was shown to be more sensitive than egg toxicity." can be better be said as. ' the result shows that the sampled water was more toxic to the sperm than the egg, or the sperm was more sensitive to the pollutants in sampled water they were exposed to than the eggs.

 Because if you say egg toxicity or sperm toxicity, it means the egg or sperm is the substance that is causing toxicity to something.  what you are studying is the toxicity of the sampled waters to egg and sperm of the animal.

Response: We respect the reviewer suggestion and throughout the manuscript the sentences with spermiotoxicity and egg toxicity were changed with appropriate words. Thanks for the reviewer suggestion.

Reviewer 3 Report

The manuscript is been revised, however, current form is still below the standard for publication. The authors need to prepare the revised manuscript more carefully and revised the main text in detail, rather than plain explain in the response letter.

Revisions should be highlighted so that reviewers can easily find the changes.

Figure 1 is still in low resolution.

There are two Figure 2s.

Presentation still need to be revised, and the reviewer suggest asking a scientific researcher who is familiar with scientific writing to go trough and edit the manuscript before resubmitting.

Author Response

Comments: The manuscript is been revised, however, current form is still below the standard for publication. The authors need to prepare the revised manuscript more carefully and revised the main text in detail, rather than plain explain in the response letter.

Revisions should be highlighted so that reviewers can easily find the changes.

Response: We carefully carried out some more revision as suggested by the reviewer and the revision was highlighted with track change.

Comments: Figure 1 is still in low resolution.

Response: As suggested by the reviewer we improved the figure 1 with high resolution and the same has been given in the revised manuscript.

Comments: There are two Figure 2s.

Response: Correction carried out

Comments: Presentation still need to be revised, and the reviewer suggest asking a scientific researcher who is familiar with scientific writing to go trough and edit the manuscript before resubmitting.

Response:  The presentation was much improved with the help of our research collaborators who is also working as an editor in Toxicology reports.

Reviewer 4 Report

The quality of the manuscript is far better. Thanks for having taking into account the suggestions.

Author Response

Thanks for reviewer

Round 3

Reviewer 3 Report

My concerns have been addressed in the revised submission, the manuscript is now acceptable.